# Estimating Semantic Alphabet Size for LLM Uncertainty Quantification

**Lucas H. McCabe**[1,2]  **Rimon Melamed**[1]  **Thomas Hartvigsen**[3]  **H. Howie Huang**[1]
[1]George Washington University  [2]LMI Consulting  [3]University of Virginia
{lucasmccabe, rmelamed, howie}@gwu.edu
hartvigsen@virginia.edu

## Abstract

Many black-box techniques for quantifying the uncertainty of large language models (LLMs) rely on repeated LLM sampling, which can be computationally expensive. Therefore, practical applicability demands reliable estimation from few samples. Semantic entropy (SE) is a popular sample-based uncertainty estimator with a discrete formulation attractive for the black-box setting. Recent extensions of SE exhibit improved LLM hallucination detection, but do so with less interpretable methods that admit additional hyperparameters. For this reason, we revisit the canonical discrete semantic entropy (DSE) estimator, finding that it underestimates the "true" semantic entropy, as expected from theory. We propose a modified semantic alphabet size estimator, and illustrate that using it to adjust DSE for sample coverage results in more accurate SE estimation in our setting of interest. Furthermore, we find that two semantic alphabet size estimators, including our proposed, flag incorrect LLM responses as well or better than many top-performing alternatives, with the added benefit of remaining highly interpretable.

## 1 Introduction

Large language models (LLMs) are not fact engines. They have been shown to forget provided context (Liu et al., 2024), fabricate records (Lee et al., 2023), and otherwise hallucinate (Ji et al., 2023). LLMs' underlying training data may be of mixed factual reliability, but models can also hallucinate when they have ample knowledge to respond adequately (Simhi et al., 2024). In risk-sensitive settings, it may be prudent for systems to abstain when uncertainty is high (or, alternatively, confidence is low) (Murphy, 2022; Hasan et al., 2025). For these reasons and others, it is prudent to estimate LLMs' intrinsic uncertainty.

Uncertainty quantification (UQ) in LLMs is particularly challenging, however, in part due to their computational scale (Liu et al., 2025). Extensive sampling can be financially prohibitive or computationally intractable, and these concerns are magnified if the UQ method is computationally complex with respect to the number of samples. Furthermore, internal activations and sequence log-probabilities are not always available from commercial inference providers (Farquhar et al., 2024), potentially disqualifying so-called "white-box" methods. Therefore, sample-efficient UQ in the black-box setting - where LLM internals are assumed inaccessible - is a crucial area of research for deploying trustworthy AI systems.

We provide empirical evidence that canonical discrete semantic entropy (DSE) underestimates the "true" semantic entropy (SE) for typical sample sizes, on average (Figure 2). To address this limitation, we first suggest a straightforward modification to existing alphabet size estimators, where we draw, in part, from population ecology. Then, we use it to adjust entropy estimates for sample coverage, resulting in reduced bias (Figure 2) and more accurate estimation of the "true" SE, compared to other black-box SE estimators (Table 1).

Finally, we evaluate UQ estimators for inaccuracy classification in sentence-length LLM responses. In our experiments, the aforementioned coverage-adjusted estimator outperforms other explicit estimators of SE (Figure 3). More strikingly, two alphabet size estimators outrank all other black-box UQ methods considered but one, after accounting for uncertainty in establishing an overall rank of

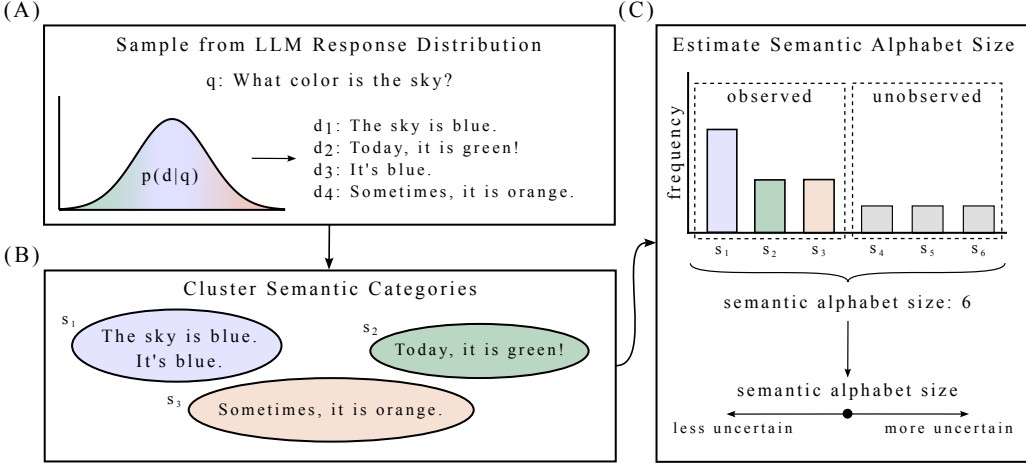

Figure 1: High-level schematic of semantic alphabet size estimation for LLM uncertainty quantification (Section 3). **(A)** Generate LLM responses to a query. **(B)** Assign responses to categories of shared meaning. **(C)** Estimate semantic alphabet size, accounting for semantic classes unobserved in the sample (Equation 9). LLM response examples are hypothetical for illustrative purposes.

methods. Our results indicate that semantic alphabet size estimation, which is highly interpretable, can perform competitively with many state-of-the art methods in LLM inaccuracy detection.

## 2 PRELIMINARIES

**Information entropy.** Information entropy quantifies the expected surprise, or uncertainty, of a random variable. For a discrete random variable $X$ with finite alphabet $S$, it is given by:

$$\mathbb{H}(X) = -\sum_{s \in S} p(s) \log p(s), \tag{1}$$

where $p(s)$ depicts the probability of symbol $s$ (Shannon, 1948). When a random variable's probability density is not known analytically, entropy estimators are employed. A well-studied approach in the finite-sample regime is the plugin approximation:

$$\hat{\mathbb{H}}_{plugin}(X) = -\sum_{i=1}^{k} \hat{p}(s_i) \log \hat{p}(s_i), \tag{2}$$

where $k$ is the number of distinct observations in a sample, and $\hat{p}(s_i)$ conveys the empirical frequency of $s_i$ among the observations.

**Semantic classification.** Lexically distinct text sequences may belong to the same *semantic equivalence class* (equivalently, "semantic category" or "semantic set") - i.e., a set of text sequences with mutually shared meaning. Sequence $d_1$ entails sequence $d_2$ if $d_1$ implies $d_2$ (Dagan et al., 2005). Kuhn et al. (2023) assign sequences $d_1$ and $d_2$ to a shared semantic equivalence class under so-called "strict entailment" if their textual entailment is bidirectional.

**Semantic entropy.** SE, introduced by Kuhn et al. (2023), aims to quantify intrinsic LLM uncertainty with a three-step procedure:

1. **Sampling**: Given a query $q$, generate $n$ passages $d_1, d_2, \ldots, d_n$.
2. **Semantic Clustering**: Iterating over pairs $(d_i, d_j)$, determine if both sequences belong to the same semantic equivalence class. Greedily assign passages to classes based on the pairwise semantic classifications.

3. **Estimation**: Semantic equivalence classes are treated as symbols in an alphabet $S$. SE is the entropy calculated over observed semantic equivalence classes:

$$SE(q|\theta) = -\sum_{s \in S} p(s|q, \theta) \log p(s|q, \theta) \tag{3}$$

where $p(s|q, \theta)$ represents the probability that an LLM parameterized by $\theta$ generates a passage belonging to semantic equivalence class $s$ in response to a query $q$.

Because the distribution over semantic sets is not known, it is approximated from response probabilities $p(d|q, \theta)$ using so-called "Rao–Blackwellized Monte Carlo integration" (RBMCI):

$$p(s_i|q, \theta) \approx \frac{\sum_d \mathbb{1}_{d \in s_i} p(d|q, \theta)}{\sum_d p(d|q, \theta)}, \tag{4}$$

where $d \in s$ indicates that response $d$ belongs to semantic equivalence class $s$, and $p(d|q, \theta)$ are response probabilities returned by the LLM, for each of the $k$ observed semantic categories $s_1, s_2, \ldots, s_k$ (Farquhar et al., 2024).

**The black box setting.** Response probabilities are not always available. Farquhar et al. (2024) consider a discrete formulation of semantic entropy (DSE), where aggregated document probabilities for each semantic equivalence class are replaced with empirical class frequencies:

$$DSE(q|\theta) = -\sum_{i=1}^{k} \left( \frac{\sum_d \mathbb{1}_{d \in s_i}}{n} \right) \log \left( \frac{\sum_d \mathbb{1}_{d \in s_i}}{n} \right) \tag{5}$$

with $n$ sampled passages and $k$ observed semantic categories. This is the plugin estimator (Equation 2) applied to semantic entropy (Equation 3). Both SE and DSE correspond with hallucination rate in question-answering problems (Farquhar et al., 2024).

**Generalizations.** Recent work is said to have generalized SE or DSE. Two pertinent examples are Kernel Language Entropy (KLE) and Semantic Nearest Neighbor Entropy (SNNE), which reported state-of-the-art performance for incorrectness classification (Nikitin et al., 2024; Nguyen et al., 2025). These stronger estimators may come at the expense of interpretability, in part owing to implementation complexity (e.g., applying a kernel to embed graph nodes in KLE) and introduction of additional hyperparameters and design choices (e.g., similarity function and scale parameter in SNNE). We elaborate on KLE and SNNE definitions and implementation details in Appendix A.5.

## 3 METHODS

The total number of semantic equivalence classes (i.e., the *semantic alphabet size* $|S|$) that may be ellicited from an LLM in response to a prompt is generally unknown, and not all categories are necessarily observed in the sample (i.e., $k < |S|$). For instance, under a simple Zipfian model of semantic category-abundance, we expect at least one category to be unobserved for sample size $n = 10$ if $|S| > 4$ (Appendix B). This situation is known as the *under-sampled regime*, where the empirical distribution over semantic categories can be less surprising than than the true one. In other words, the plugin method for DSE may underestimate LLM uncertainty, which we illustrate in Figure 2. For this reason, we are interested in methods for estimating semantic alphabet size and adjusting for it when estimating SE.

**Semantic alphabet size.** Because plugin DSE does not directly account for unobserved semantic categories, the implicit alphabet size used by the plugin estimator is $k$, called "NumSets" by Lin et al. (2024). In the under-sampled regime, NumSets underestimates $|S|$, so a more accurate semantic alphabet size estimator for small $n$ is desirable. Parallels exist with the so-called "unseen species" problem in population ecology: given a sample of $n$ observations belonging to one or more species, estimate the number of yet unseen species that would be discovered by collecting additional observations (Fisher et al., 1943). In this setting, the sample coverage $C$, the fraction of all possible categories observed in a sample (i.e., $C = \frac{k}{|S|}$), is also of interest (Chao & Shen, 2003).

The so-called "Good-Turing" sample coverage estimator is $\hat{C}_{GT} = 1 - \frac{f_1}{n}$, where $f_1$ is the number of singletons, or semantic categories observed only once (Good, 1953). Modest arithmetic converts the Good-Turing sample coverage estimator into an alphabet size estimator:

$$\widehat{|S|}_{GT} = \frac{kn}{n - f_1}. \tag{6}$$

More recently, Lin et al. (2024) suggested a "continuous" NumSets analogue: responses are interpreted as nodes of a fully-connected graph $G$ with weights $w_{ij} = \frac{a_{ij} + a_{ji}}{2}$, where $a_{ij}$ is the entailment probability for response pair $d_i$, $d_j$, via an NLI model. Given the eigenvalues $(\lambda_1 < \cdots < \lambda_n)$ of $G$'s normalized Laplacian, the estimator is given by:

$$U_{EigV} = \sum_{i=1}^{n} \max(0, 1 - \lambda_i). \tag{7}$$

**Coverage-adjusted entropy.** To quantify a population's ecological diversity, Chao & Shen (2003) provide a coverage-adjusted discrete entropy estimator that scales empirical category frequencies by estimated sample coverage:

$$\hat{\mathbb{H}}_{CS} = -\sum_{i=1}^{k} \frac{\hat{C}_{GT} \hat{p}_i \log(\hat{C}_{GT} \hat{p}_i)}{1 - (1 - (\hat{C}_{GT} \hat{p}_i))^n}, \tag{8}$$

where $\hat{p}_i = \frac{1}{n} \sum_d \mathbb{1}_{d \in s_i}$. This so-called "Chao-Shen" estimator is consistent and less biased than many empirical alternatives (Vu et al., 2007; Pinchas et al., 2024).

**Hybrid estimators.** First, we adapt the two aforementioned semantic alphabet size estimators to address shortcomings of each. When the number of singletons is zero, $\widehat{|S|}_{GT}$ reduces to NumSets, and it is undefined when all samples belong to distinct semantic categories. On the other hand, $U_{EigV}$ can be less than $k$, which is a lower bound for $|S|$. To remediate the above limitations, we propose an alternative "hybrid" semantic alphabet size estimator:

$$\widehat{|S|}_{Hybrid} = \begin{cases} U_{EigV}, & \text{if } f_1 = n \\ max\left(\widehat{|S|}_{GT}, U_{EigV}\right), & \text{otherwise.} \end{cases} \tag{9}$$

Additionally, we propose a Chao-Shen-like DSE estimator that takes the form of Equation 8, but invokes the hybrid semantic alphabet size estimator for coverage adjustment:

$$\hat{\mathbb{H}}_{Hybrid} = -\sum_{i=1}^{k} \frac{\frac{k\hat{p}_i}{\widehat{|S|}_{Hybrid}} \log\left(\frac{k\hat{p}_i}{\widehat{|S|}_{Hybrid}}\right)}{1 - \left(1 - \frac{k\hat{p}_i}{\widehat{|S|}_{Hybrid}}\right)^n}. \tag{10}$$

## 4 EXPERIMENTS

### 4.1 EXPERIMENTAL SETTINGS

**Models.** Following Farquhar et al. (2024), we focus on fine-tuned models - in our case, the instruction-tuned models of Gemma-2-9B (Team et al., 2024), Gemma-3-12B (Team et al., 2025), Llama-3.1-8B (Grattafiori et al., 2024), Mistral-v0.3-7B, and Phi-3.5-3.8B (Abdin et al., 2024). We perform text generation at two temperatures, for distinct purposes. Following related prior work, we sample at temperature $\tau = 1.0$ to calculate uncertainty scores and again at $\tau = 0.1$ to obtain a "best guess" response for assessing the correctness of LLM responses (Farquhar et al., 2024; Nikitin et al., 2024; Nguyen et al., 2025). With the exception of Figure 2, which iterates over several sample sizes, the results shown in the main body of this work use a sample size of $n = 10$, which is also found in prior work on SE (Kuhn et al., 2023; Farquhar et al., 2024; Nikitin et al., 2024; Nguyen et al., 2025). Due to its reduced computational burden compared to larger sample sizes and prevalence in the relevant literature, we will refer to $n = 10$ as a "practical" or "typical" sample size.

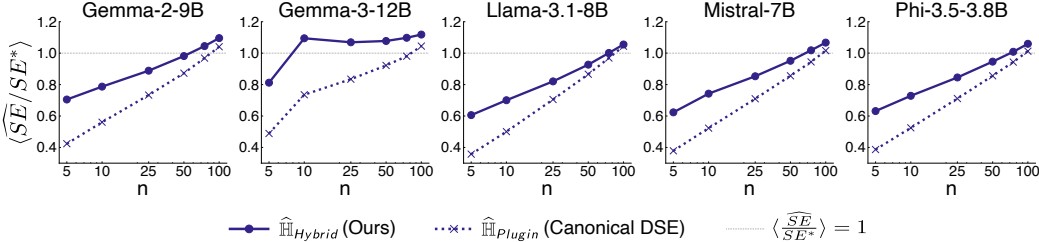

Figure 2: Illustrating underestimation in discrete semantic entropy (DSE) calculation with typical sample sizes. The ratios of DSE estimators with varying sample size ($n = 5, 10, 25, 50, 75, 100$) to white-box SE with $n = 100$ (denoted $SE^*$) are shown, with values below 1 suggesting underestimation (dotted grey line). The estimators displayed are the plugin estimator of canonical DSE (i.e., Equation 5, dotted indigo line) and the "hybrid" DSE estimator of Equation 10 (solid indigo line). Results are averaged across queries within each dataset, then uniformly averaged across datasets. Log scale is used on the x-axis to highlight differences between estimators with smaller sample sizes. Instances with a denominator of 0 are ignored.

**Datasets.** We consider question-answering datasets wherein each query has either one or multiple correct answer(s). For the former, we use the validation sets of HotpotQA (Yang et al., 2018) and SQuAD 2.0 (Rajpurkar et al., 2018; 2016), which contain 7405 and 11864 exemplars, respectively. Our experiments require extensive sampling of LLM responses, and BEC demands combinatorially many language model calls. For these reasons, and because SQuAD 2.0 is much larger than the other data sets, our experiments using it are performed on a random 20% subset. HotpotQA has substantial scale, as well, due to its inclusion of contextual information supplementing the queries. For estimator evaluation (Figure 2 and Table 1), we use the first 25% of HotpotQA's queries; for incorrectness detection (Figure 3), we use the full dataset. We also consider queries with a wider variety of possible correct answers. The first such dataset is BioASQ, a biomedical QA benchmark with 4719 exemplars. Each question in BioASQ has between 1 and 15 reference answers. We also prepare a small (131 exemplars) supplementary dataset called Plethora Of accepTable cATegOries (POTATO), for which the number of possible correct semantic categories has an even larger range (up to 722). We discuss POTATO in detail in Section A.2.

## 4.2 METRICS AND BASELINES

**Semantic entropy estimator evaluation.** For a given prompt-LLM pair, $|S|$ and the true probability distribution over semantic equivalence classes are unknown, so semantic alphabet size and SE have no ground truths. Instead, we assume that white-box SE with a large number of samples ($n = 100$), denoted $SE^*$, is interchangeable with the true estimand (i.e., $SE^*$ well-represents the true entropy over all possible semantic categories). We elaborate on this choice in Appendix Section C. When illustrating the degree to which $SE^*$ is underestimated (i.e., Figure 2), we report the ratio of the estimator of interest to $SE^*$. Since information entropy is non-negative, values below 1 indicate underestimation. Our primary evaluation metric for entropy estimators, however, is mean-squared error (MSE): we assess the MSE between an estimate using the small sample size ($n = 10$) and $SE^*$.

**Incorrectness evaluation.** We also assess the ability of estimators to classify LLM responses as "correct" or "incorrect," sometimes referred to as "hallucination" or "confabulation" detection (Farquhar et al., 2024). We calculate the area under the receiver operating characteristic curve (AUROC), corresponding to the probability that a randomly selected incorrect LLM response has a higher uncertainty score than a randomly selected correct response. Empirical AUROC values are point estimates, which themselves have uncertainty. Nikitin et al. (2024) point out that such uncertainty is strongly driven by the model and dataset, rather than UQ method, motivating head-to-head comparisons and evaluation by win rate. We advance a similar approach, but we rely on Bradley-Terry latent strength scores (Zermelo, 1929; Bradley & Terry, 1952), calculated via minorization-maximization (Caron & Doucet, 2012; Hunter, 2004), allowing us to establish an overall rank of methods. We also employ a Monte Carlo procedure that attempts to account for uncertainty in estimating both

| Dataset | Estimator | Gemma-2 (9B) | Gemma-3 (12B) | Llama-3.1 (8B) | Mistral-v0.3 (7B) | Phi-3.5 (3.8B) |
|---|---|---|---|---|---|---|
| HotpotQA | $\widehat{\mathbb{H}}_{Plugin}$ | $0.46 \pm 0.02$ | $0.09 \pm 0.01$ | $0.68 \pm 0.03$ | $0.59 \pm 0.03$ | $0.61 \pm 0.03$ |
| | $\widehat{\mathbb{H}}_{CS-GT}$ | $0.39 \pm 0.02$ | $0.08 \pm 0.01$ | $0.56 \pm 0.03$ | $0.46 \pm 0.03$ | $0.47 \pm 0.03$ |
| | $\widehat{\mathbb{H}}_{Hybrid}$ | $\mathbf{0.30} \pm 0.02$ | $\mathbf{0.06} \pm 0.01$ | $\mathbf{0.45} \pm 0.02$ | $\mathbf{0.39} \pm 0.02$ | $\mathbf{0.39} \pm 0.02$ |
| SQuAD 2.0 | $\widehat{\mathbb{H}}_{Plugin}$ | $0.68 \pm 0.03$ | $0.18 \pm 0.02$ | $0.78 \pm 0.03$ | $1.37 \pm 0.04$ | $1.43 \pm 0.04$ |
| | $\widehat{\mathbb{H}}_{CS-GT}$ | $0.51 \pm 0.03$ | $0.15 \pm 0.01$ | $0.60 \pm 0.03$ | $0.86 \pm 0.06$ | $0.91 \pm 0.06$ |
| | $\widehat{\mathbb{H}}_{Hybrid}$ | $\mathbf{0.43} \pm 0.02$ | $\mathbf{0.12} \pm 0.01$ | $\mathbf{0.50} \pm 0.02$ | $\mathbf{0.68} \pm 0.03$ | $\mathbf{0.75} \pm 0.03$ |
| POTATO | $\widehat{\mathbb{H}}_{Plugin}$ | $0.39 \pm 0.08$ | $0.35 \pm 0.08$ | $0.62 \pm 0.11$ | $1.71 \pm 0.15$ | $1.91 \pm 0.15$ |
| | $\widehat{\mathbb{H}}_{CS-GT}$ | $0.30 \pm 0.07$ | $0.29 \pm 0.07$ | $0.51 \pm 0.11$ | $0.94 \pm 0.29$ | $1.58 \pm 0.55$ |
| | $\widehat{\mathbb{H}}_{Hybrid}$ | $\mathbf{0.27} \pm 0.06$ | $\mathbf{0.24} \pm 0.06$ | $\mathbf{0.42} \pm 0.09$ | $\mathbf{0.71} \pm 0.11$ | $\mathbf{0.73} \pm 0.11$ |
| BioASQ | $\widehat{\mathbb{H}}_{Plugin}$ | $1.64 \pm 0.03$ | $1.79 \pm 0.03$ | $1.68 \pm 0.03$ | $2.04 \pm 0.03$ | $1.82 \pm 0.02$ |
| | $\widehat{\mathbb{H}}_{CS-GT}$ | $0.96 \pm 0.04$ | $1.10 \pm 0.07$ | $1.06 \pm 0.05$ | $1.31 \pm 0.08$ | $0.99 \pm 0.04$ |
| | $\widehat{\mathbb{H}}_{Hybrid}$ | $\mathbf{0.78} \pm 0.02$ | $\mathbf{0.69} \pm 0.02$ | $\mathbf{0.80} \pm 0.02$ | $\mathbf{0.73} \pm 0.02$ | $\mathbf{0.83} \pm 0.02$ |

Table 1: Empirically evaluating the accuracy of explicit discrete semantic entropy (DSE) estimators. Values reflect MSE between the estimated value using $n = 10$ samples and white-box semantic entropy with $n = 100$ (i.e., $SE^*$). The lowest MSE value for each model-dataset pair is shown in bold. Intervals, shown in grey, reflect 95% CIs via the standard error of the mean.

AUROC and strength scores. First, we fit Gaussian uncertainty distributions about AUROC point estimates from the 95% confidence intervals (CIs) obtained via DeLong's method (DeLong et al., 1988; Sun & Xu, 2014), a modeling assumption we justify by the approximate normality of the U statistic (Mann & Whitney, 1947).[1] For each model-dataset pair, we simulate $L = 100$ matches between each pair of methods by comparing AUROC values sampled from their corresponding uncertainty distributions. After obtaining strength scores from the simulated matches, we calculate CIs about the Bradley-Terry scores and establish 95% CIs about the ranks of latent strengths using the methods of Gao et al. (2023). We provide additional details in Appendix A.6.

**Baselines.** Because our priority is black-box uncertainty estimation, we compare with other black-box methods. For SE estimation, we compare our hybrid approach (Equation 10) to the other explicit black-box SE estimators: the canonical discrete approach of Farquhar et al. (2024) (the plugin estimator, Equation 5) and the coverage-adjusted estimator of Chao & Shen (2003) (Equation 8). For incorrectness classification, we consider the three aforementioned explicit black-box SE estimators, four alphabet size estimators, and four other uncertainty methods. The alphabet size estimators are the number of semantic categories (i.e., NumSets) (Kuhn et al., 2023), the alphabet size estimator converted from the Good-Turing sample coverage estimator (Good, 1953), the spectral estimator of Lin et al. (2024) (i.e., $U_{EigV}$), and our hybrid alphabet size estimator (i.e., Equation 9). The other uncertainty methods are Predictive Entropy (PE, see Appendix A.5) (Kadavath et al., 2022), SNNE (Nguyen et al., 2025), KLE (Nikitin et al., 2024), and white-box SE (Kuhn et al., 2023).

### 4.3 RESULTS

**Canonical discrete semantic entropy underestimates "true" semantic entropy.** The plugin estimator for information entropy $\hat{\mathbb{H}}_{plugin}$ has a theoretically-established negative bias (Basharin, 1959; Harris, 1975), which largely governs its mean-squared error (Antos & Kontoyiannis, 2001). We are interested in observing this phenomenon empirically, where we do not have access to the true distribution over semantic equivalence classes. In Figure 2, we compare plugin DSE estimates to $SE^*$, indicating that the canonical plugin DSE approach underestimates its quantity of interest for practical sample sizes. Such underestimation can be problematic for reliable UQ, because drawing a large number of samples from an LLM can be costly and time-consuming at scale.

---

[1]In principle, samples from these distributions may exist outside the [0, 1] range, but this is unlikely in our case, given the AUROC CI bounds (Figure 10).

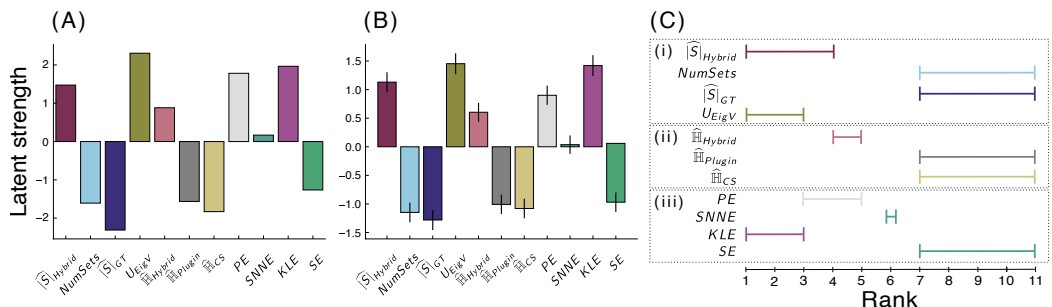

Figure 3: Establishing overall performance of ten UQ methods on incorrectness detection. **(A)** Bradley-Terry latent strength scores from pairwise comparison of AUROC point estimates. **(B)** Bradley-Terry latent strength scores after accounting for uncertainty in estimating AUROC. Error bars are "conservative" CIs about strength scores, which may be slightly stricter than 95%; see Section A.6 for details. **(C)** For each method, we establish 95% CIs about the rank of Bradley-Terry latent strength MLEs (Gao et al., 2023) for the incorrectness detection task; see Section A.6 for details. We highlight **(i)** semantic alphabet size estimators, **(ii)** black-box discrete semantic entropy (DSE) estimators, and **(iii)** other uncertainty estimators, which include white-box (PE, SE) and black-box methods (SNNE, KLE). The interval [a, b] denotes all integers from $a$ to $b$, inclusively. The CI for SNNE reflects [6, 6] and is extended for readability.

**Our proposed estimator improves the accuracy of discrete semantic entropy estimation.** In Figure 2, we observe that, on average, $\hat{\mathbb{H}}_{Hybrid}$ misestimates its target quantity by less than canonical plugin DSE for a range of sample sizes. We perform a more granular comparison of SE estimator accuracy with $n = 10$ in Table 1, where $\hat{\mathbb{H}}_{Hybrid}$ consistently achieves the lowest MSE among explicit discrete SE estimators across five models and four datasets. We also find that $\hat{\mathbb{H}}_{Hybrid}$ slightly *over*-shoots $SE^*$ on average for the 12B-parameter Gemma model. Detailed results, broken out by dataset in Appendix Figure 7, reveal that this overshooting is limited to the POTATO dataset. In fact, this appears to be driven by just three anomalous instances, examined in Appendix D. Excluding instances with near-zero "true" entropy (i.e., $SE^* < 0.005$) recovers the pattern exhibited by the other models (Figure 9).

**Our proposed semantic entropy estimator outperforms other explicit semantic entropy estimators at incorrectness detection.** LLM uncertainty estimation is often employed as a proxy for hallucination, confabulation, or incorrectness detection among LLM responses (Kuhn et al., 2023; Farquhar et al., 2024). Following Nikitin et al. (2024), we calculate pairwise AUROC win rates between models – for each model-dataset pair, a win is recorded in method $m_i$'s favor over method $m_j$ if the corresponding AUROC for $m_i$ is greater than that of $m_j$ (Figure 4). We characterize overall performance of each method via Bradley-Terry latent strength scores. Figure 3A displays maximum likelihood estimates of the methods' strength scores from pairwise AUROC comparisons. We highlight the uncertainty in both AUROC and Bradley-Terry estimation in Figure 3B (CIs about Bradley-Terry estimates, which may be slightly stricter than 95%) and Figure 3C (95% CIs around each method's rank by Bradley-Terry score). Here, $\widehat{H}|_{Hybrid}$ (Equation 10) achieves the highest rank among explicit estimators of semantic entropy, including white-box SE.

**Alphabet size estimators alone can outperform entropy estimators at incorrectness detection.** After accounting for uncertainty in our rank-generating procedure, two semantic alphabet size estimators - namely, $\widehat{|S|}_{Hybrid}$ and $U_{EigV}$ - outperform many more complex black-box uncertainty methods. Three methods have 95% rank CIs that include the top position: alphabet size estimators $\widehat{|S|}_{Hybrid}$ (rank CI of [1, 4]) and $U_{EigV}$ (rank CI of [1, 3], which attains the largest latent strength estimate), as well as KLE (rank CI of [1, 3]) (Figure 3C), with all three methods outperforming white-box SE and other explicit SE estimators overall. Our results recall Kuhn et al. (2023)'s observation that the number of observed semantic categories is itself "a reasonable uncertainty measure."

## 5 RELATED WORK

Herein, we briefly review related contributions not discussed elsewhere in this work.

**Earlier LLM uncertainty quantification methods.** Linguistic Confidence assesses whether the LLM articulates its confidence in its response (Mielke et al., 2022). The P(True) method of Kadavath et al. (2022) similarly relies on an LLM's self-perceived uncertainty, asking a language model if a response is "True" or "False," with the response probability of "False" ultimately reported. SelfCheckGPT draws $n + 1$ responses from an LLM and assesses the consistency of each sentence of the first response with each of the $n$ following responses (Manakul et al., 2023). In prior work, such approaches have been consistently superseded by the methods otherwise described herein (Kuhn et al., 2023; Lin et al., 2024; Farquhar et al., 2024; Nikitin et al., 2024; Nguyen et al., 2025).

**Uncertainty and internal representations.** In the white-box setting, it may be desirable to ascertain LLM uncertainty from internal representations, rather than repeated sampling. Han et al. (2024b) and Han et al. (2024a), for example, approach this from an interpretability point of view, building on earlier work on semantic entropy (Kuhn et al., 2023; Farquhar et al., 2024). Other analyses, however, contend that so-called "truthfulness encodings" are difficult to generalize (Orgad et al., 2025), warranting further examination.

**Alphabet size estimation.** The Good-Turing estimator discussed herein accounts for singletons in the sample (Chao & Shen, 2003). Alternative alphabet size estimators accounting for doubletons (i.e., semantic categories appearing exactly twice in the sample) (Chao, 1987) and tripletons (i.e., categories appearing exactly three times) (Lanumteang & Böhning, 2011) exist, but they are undefined when the number of doubletons is zero.

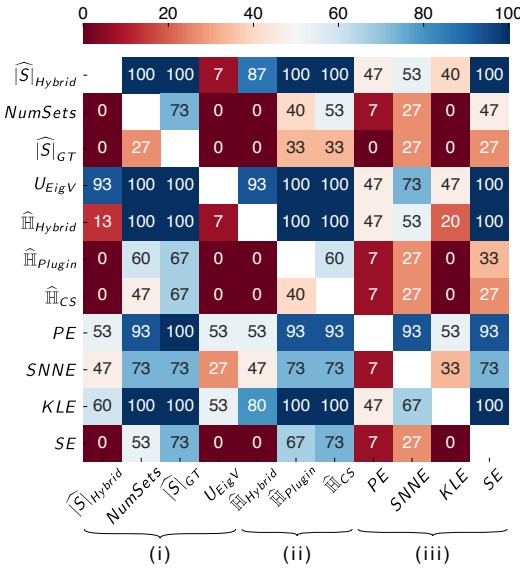

Figure 4: Heatmap illustrating the proportion of model-dataset pairs, rounded to the nearest integer, for which a row's method achieved a larger AUROC point estimate than a column's method. Uncertainty methods are organized into three groups: **(i)** semantic alphabet size estimators, **(ii)** black-box discrete semantic entropy (DSE) estimators, and **(iii)** other uncertainty estimators, which include white-box (PE, SE) and black-box methods (SNNE, KLE). The hybrid discrete semantic entropy (DSE) estimator consistently outperforms other explicit SE estimators.

**Unbiased entropy estimation.** The entropy estimator of Montgomery-Smith & Schürmann (2014) is unbiased, but it is incompatible with the possibility of unobserved semantic equivalence classes. Otherwise, no unbiased estimator exists, to the best of our knowledge (Paninski, 2003).

## 6 DISCUSSION

### 6.1 CONCLUSION

Several approaches for estimating LLM uncertainty have emerged in recent years, but UQ performance can be constrained, absent white-box LLM access. Furthermore, the practicality of sampling-based UQ methods is limited by the computational costs associated with repeated text generation. We illustrate the importance of semantic alphabet size in LLM uncertainty estimation with small sample sizes. Adjusting DSE for sample coverage using our proposed semantic alphabet size estimator results in more accurate SE estimation and improved LLM incorrectness classification, com-

pared to other explicit estimators of SE. Further, we find that semantic alphabet size estimators $U_{EigV}$ and $\widehat{|S|}_{Hybrid}$ achieve higher latent strength estimates than most UQ methods considered in our incorrectness detection experiments, with $U_{EigV}$ sharing the top rank CI with KLE, a conceptually complex approach that may be more difficult to interpret.

The overall performance of SE and plugin DSE on the incorrectness task are qualitatively similar (Figure 3), and related findings appear in prior work, such as Farquhar et al. (2024) and Nguyen et al. (2025). In fact, our experiments indicate that white-box SE tends to track plugin DSE quite closely, on average, as seen in Figure 9. We reflect on this phenomenon in Appendix Section E, where we observe nearly identical distributions induced by RBMCI and empirical class frequencies, on average. A notable difference is that SE appears to be more robust to instances where exceedingly rare semantic categories are overrepresented in the sample. The sequence log-probabilities available to SE may be otherwise underutilized, however, since SE exhibits the same underestimation pattern as plugin DSE. Future work may extend the approach described herein to the white-box setting by directly adjusting SE for unobserved semantic categories.

Though out of scope for the present study, we underscore that semantic alphabet size estimation - and $\widehat{|S|}_{Hybrid}$, in particular - may have broader application than merely entropy estimation. For instance, concurrent work by Li et al. (2025) applied a Good-Turing method to estimate the extent of LLMs' unexpressed factual knowledge (e.g., mathematical theorems and diseases).

### 6.2 Limitations

We highlight several limitations in the present work. First, computational constraints limit our ability to run the experiments herein with models larger than 12 billion parameters. That said, our analysis is extensive, spanning five models, four datasets, and eleven uncertainty estimators. Though our results are empirical, and experimental results may vary across studies, we make efforts to express our statistical uncertainty, and our final rank-generating procedure aims to account for it.

Second, our work aims to improve upon plugin DSE, whose estimator is negatively biased. For this reason, we take semantic cluster labels as fixed, without ablating across alternative clustering strategies. End-to-end SE calculation involves several steps, however, and a biased estimator does not necessitate that the entire process results in a negatively biased estimate. For instance, a high false negative rate in the assignment of responses to semantic categories (e.g., due to minor lexical alterations inducing false negatives) may positively bias the final estimate (Grewal et al., 2024).

Finally, the uncertainty estimation methods considered herein do not directly measure factual inaccuracy. Instead, they may be better understood as indicators of semantic diversity. Of course, high semantic diversity may correspond to a high factual error rate, but there are scenarios wherein semantic diversity does not necessarily imply incorrectness (Ilia & Aziz, 2024), and, conversely, models can be confidently wrong.

### 6.3 Reproducibility

We outline our major experimental settings in Section 4.1, as well as evaluation metrics and baseline methods in Section 4.2. In Appendix A, we elaborate extensively on additional implementation details needed to reproduce our results, including algorithms, models, and prompt templates. Interested readers will also find a brief overview of implementation variations found in other work in Section H. Finally, we provide a code repository at https://github.com/lucasmccabe/esa, with hope to support scientific accessibility and reproducibility in future work.

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

## A  ADDITIONAL IMPLEMENTATION DETAILS

### A.1  TEXT GENERATION

Throughout this work, we generate a maximum of 100 tokens for each response, with early stopping enabled if the EOS token is generated. All otherwise unspecified hyperparameters are set to their defaults (e.g., we do not invoke top-$k$ sampling, top-$p$ sampling, beam search, repetition penalty, etc.). For incorrectness classification, we follow prior work that elicits concise responses by prepending QA queries with the following "pre-prompt" (Farquhar et al., 2024; Nikitin et al., 2024):

> **Pre-Prompt for Single-Sentence QA**
>
> Answer the following question in a single brief but complete sentence.

## A.2 SYNTHETIC DATA GENERATION

Plethora Of accepTable cATegOries (POTATO) is a small synthetic dataset wherein every question is, in principle, answerable with a one-word or few-word response, and each question has more than one correct answer. We repeatedly invoke the following prompt against OpenAI's *GPT-4-turbo*:

> **Prompt for POTATO Question Generation**
>
> Using the available function, generate 10 questions from diverse topic areas. Each question should request only a single answer, but there should be exactly [NUM_ANSWERS] possible semantically distinct correct answer(s) to the question. For example, 'Name a continent on Earth' has seven possible correct answers, because Earth has seven continents, but 'Name all the continents on Earth' only has one possible semantically distinct correct answer (a list of all seven continents).

Above, [NUM_ANSWERS] is an integer between 1 and 50, and the available function is a Pydantic object enforcing structured generation (Colvin et al., 2024). A human annotator assessed the number of semantic categories for possible correct answers, discarded those with only one possible correct semantic category, and removed duplicate questions. The resulting dataset has 131 unique questions. We also generate 100 responses to each question from the POTATO dataset using *GPT-4o-mini*. A human annotator manually assigned the responses to semantic categories.

One query ("Name a piece of classical music composed by Ludwig van Beethoven.") has a particularly large number of possible correct semantic categories, owing to Beethoven's prolific output of several hundred compositions. Figure 5A suggests that the distribution of semantic alphabet sizes for model-query pairs is generally dominated by the distribution of the numbers of possible correct semantic categories for those queries. On a per-query basis, however, these quantities exhibit low correlation (Figure 5B).

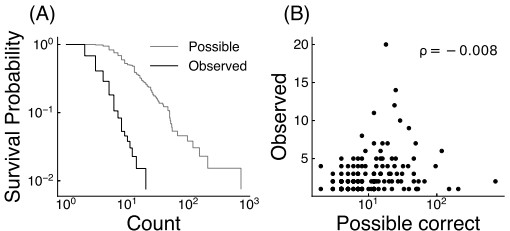

Figure 5: **(A)** Empirical survival function of the number of possible correct semantic categories and the number of observed semantic categories in the responses generated by GPT-4o-mini, according to a human annotator. **(B)** Scattergram of the number of observed semantic categories ("Observed"), according to the human annotator, against the number of possible correct semantic categories ("Possible"). One question is excluded ("Identify a programming language designed by Microsoft."), because the total number of possible correct semantic categories was not known by the authors of this work.

## A.3 BIDIRECTIONAL ENTAILMENT CLUSTERING

**Semantic classification.** The semantic clustering step of semantic entropy calculation is performed using a Bidirectional Entailment Clustering (BEC), which classifies pairs of passages $(d_i, d_j)$ bidirectionally as "entailment," "neutral," or "contradiction," and uses these labels to greedily assign passages to semantic equivalence classes. Like previous work, our implementation focuses on strict entailment, where both unidirectional relations must be considered "entailment" or equivalent (i.e., not "neutral"). We refer the reader to Kuhn et al. (2023); Farquhar et al. (2024) for further detail on the BEC algorithm. The classifier used varies by implementation, either an LLM- or NLI-based method:

| Method | Variant | $|S| > 1$ | | | $|S| = 1$ | | |
|---|---|---|---|---|---|---|---|
| | | FMI | NMI | PA | FMI | NMI | PA |
| Bidirectional Entailment | LLM | 0.84 | **0.72** | 0.84 | 0.86 | **0.71** | 0.80 |
| | NLI-NQ | 0.84 | 0.64 | **0.85** | **0.89** | 0.61 | **0.89** |
| Semantic Embedding | HDBSCAN | 0.68 | 0.52 | 0.69 | 0.70 | 0.41 | 0.60 |
| | DBSCAN ($\epsilon = 0.05$) | 0.86 | 0.71 | 0.82 | 0.86 | 0.65 | 0.77 |
| | DBSCAN ($\epsilon = 0.075$) | **0.87** | 0.68 | 0.82 | 0.88 | 0.60 | 0.79 |
| | DBSCAN ($\epsilon = 0.1$) | 0.85 | 0.57 | 0.79 | 0.88 | 0.51 | 0.80 |
| | DBSCAN ($\epsilon = 0.15$) | 0.79 | 0.27 | 0.66 | 0.87 | 0.25 | 0.76 |

Table 2: Evaluating semantic clustering methods against human-annotated ground truth. Results are averaged across questions from the POTATO dataset, either excluding questions for which all responses were semantically equivalent or questions eliciting multiple semantic categories ($|S| > 1$ and $|S| = 1$, respectively), according to the human rater. Metrics considered are the Fowlkes–Mallows index (FMI), normalized mutual information (NMI), and pairwise agreement (PA), assessed between the semantic equivalence classes assigned by the provided method and the human-annotated ground truth. Values are rounded to two decimal places. The best-performing method by each metric is in bold. Responses are generated by *GPT-4o-mini*.

**Large Language Model (LLM).** Farquhar et al. (2024) classify passage pairs (T1, T2) by invoking OpenAI's *GPT-3.5* endpoint with the following prompt:

> **Prompt for Entailment Classification**
>
> We are evaluating answers to the question {question}
> Here are two possible answers:
> Possible Answer 1: {T1}
> Possible Answer 2: {T2}
> Does Possible Answer 1 semantically entail Possible Answer 2? Respond with entailment, contradiction, or neutral.

Our implementation invokes OpenAI's *GPT-4o-mini* model, due to the sunsetting of *GPT-3.5*.

**Natural Language Inference (NLI).** The BEC approach of Kuhn et al. (2023) classifies $(d_i, d_j)$ using a *DeBERTa* model fine-tuned for NLI tasks.[2] Our implementation, used throughout this work, updates the classifier to a fine-tuned NLI model based on Microsoft's *DeBERTaV3* (He et al., 2021).[3] The NLI classification can be performed with or without the source questions (NLI-Q and NLI-NQ, respectively); in the former, the query is prepended to both $d_i$ and $d_j$ before passing through the NLI model.

**Comparison.** To compare the alignment between each clustering method and the human-annotated ground-truth, we measure the Fowlkes–Mallows index (FMI) (Fowlkes & Mallows, 1983) and normalized mutual information (NMI) (Danon et al., 2005; Lancichinetti et al., 2009). We also consider pairwise agreement (PA), wherein we iterate over pairs of passages, classify each pair as "entailment" or "contradiction," based on the clustering results, and report for each method the portion of pairs whose entailment label agrees with that of the human annotator (similar to the evaluation done by Farquhar et al. (2024)). For some queries, the human annotations assign all responses to the same semantic equivalence class (i.e., $|S| = 1$). To ensure clustering methods are performant in both $|S| = 1$ and $|S| > 1$ scenarios, we calculate metrics for each (Table 2). Although this paper inherits its BEC semantic clustering strategy from prior work (Kuhn et al., 2023; Farquhar et al., 2024), we also consider several comparative baselines based on semantic embedding distance. In particular, we embed responses using *bge-base-en-v1.5*[4] and cluster according to their cosine distances via either density-based spatial clustering of applications with noise (DBSCAN) (Ester et al., 1996) with

---

[2] https://huggingface.co/microsoft/deberta-large-mnli
[3] https://huggingface.co/cross-encoder/nli-deberta-v3-base
[4] https://huggingface.co/BAAI/bge-base-en-v1.5

radius parameter $\epsilon = 0.05, 0.075, 0.1, 0.15$ or hierarchical DBSCAN (HDBSCAN) (Campello et al., 2013).

By most metrics, the best alignment with the human labels in each scenario (i.e., $|S| = 1$ vs. $|S| > 1$) is achieved by a BEC-based strategy (Table 2). The baselines using semantic embeddings have the additional limitation of relying on semantic similarity as a proxy for semantic equivalence. Among BEC strategies, the NLI variant without the inclusion of questions performs roughly similarly to LLM-based alternative. For its comparable performance while limiting costs, we use NLI-NQ throughout this work.

## A.4   LLM-AS-JUDGE

Prior work labeled an LLM's response as correct if the ROUGE-L score (Lin, 2004; Lin & Och, 2004) between the response and a reference answer was above 0.3 (Kuhn et al., 2023). Appropriate thresholds may vary by model and dataset - for instance, the reference answers in HotPotQA appear rather brief, and models can vary in their verbosity, without it necessarily impacting the correctness of their responses. Additionally, such methods may not capture semantically equivalent, but lexically distinct, paraphrasings (e.g., using an acronym).

Instead, Lin et al. (2024) prompt a commercial LLM to provide a numerical rating of consistency between LLM responses and reference answers, with ratings above 70 indicating an accurate response. They observe, however, that a small portion of LLM judgements do not have an easily-parseable rating. For this reason, we modify their prompt to request ratings within XML-style tags and include an in-context example:

> **Prompt for Consistency-Based Judge**
>
> Rate the level of consistency between the answer to the question and the reference answer, from 0 to 100. Output the float rating inside <rating></rating> tags.
> Here is an example output:
> Question: What is the capital of France? Reference: Paris is the capital of France. Answer: The capital of France is Paris. <rating>100</rating>
> Now rate the following:
> Question: {question}
> Reference: {groundtruth}
> Answer: {pred}

When a query has multiple reference answers (e.g., those in the BioASQ), we invoke the above prompt to compare the LLM response to each reference answer, then report the maximum LLM-as-judge rating. Our choice of commercial model for LLM-as-judge is OpenAI's *GPT-4o-mini*.

## A.5   OTHER UNCERTAINTY ESTIMATORS

**Predictive entropy.** The so-called "total uncertainty" associated with a prediction is the information entropy of the corresponding predictive posterior distribution (Malinin & Gales, 2021). In UQ for natural language generation, an analogue is predictive entropy (PE), the entropy of the distribution of generable sequences by an LLM in response to a prompt (Kuhn et al., 2023; Lin et al., 2024; Farquhar et al., 2024). Since not all generable sequences are typically known, PE is calculated using the plugin method to estimate the entropy of a prompt-model pair's "answer distribution" (Kadavath et al., 2022). A shortcoming of this approach is that it treats all lexically distinct sequences as unique elements of an alphabet, even if they are semantically similar.

**SNNE.** SNNE aims to account for both "intra-and inter-cluster similarity." Provided a scale factor $\tau$ and a similarity function $f$ for assessing passage pairs $(d_i, d_j)$, SNNE is defined as:

$$SNNE(q) = -\frac{1}{n} \sum_{i=1}^{n} \log \sum_{j=1}^{n} \exp\left(\frac{f(d_i, d_j)}{\tau}\right) \tag{11}$$

(Nguyen et al., 2025). Our implementation uses ROUGE-L for $f$ and scale factor $\tau = 1$, which the authors report as best-performing (Nguyen et al., 2025).

**KLE.** Like $U_{EigV}$, KLE constructs a graph over LLM responses to a query, where nodes are responses and edge weights are prescribed by the results of an NLI model. Instead of using the model's categorical scores, however, KLE's semantic graph has weights $w_{i,j} = g(d_i, d_j) + g(d_j, d_i)$, where $g$ is 1 if the NLI model predicts "entailment" for the provided response pair, 0.5 if it predicts "neutral," and 0 otherwise. The standard graph Laplacian (i.e., the difference between the degree matrix and the weight matrix) is then taken over the resultant semantic graph (Nikitin et al., 2024).

KLE admits a choice of kernel to apply to the Laplacian, resulting in a "density" matrix $K$. Given the eigenvalues ($\lambda_1 < \cdots < \lambda_n$) of $K$, the von Neumann entropy is calculated:

$$VNE(K) = \sum_{i=1}^{n} \lambda_i \log \lambda_i. \tag{12}$$

Although the concept of von Neumann entropy has its roots in quantum physics (Petz, 2001), it has also been used more recently in network science literature to quantify graph complexity (Minello et al., 2019). Nikitin et al. (2024) report the heat kernel as best-performing overall:

$$K_{heat} = e^{-tL}. \tag{13}$$

Our implementation of KLE invokes $K_{heat}$ with hyperparameter $t = 0.3$, which the authors consider a "reasonable default" that outperforms prior approaches without additional hyperparameter optimization (Nikitin et al., 2024).

**Explicit estimation of semantic entropy.** We do not include SNNE or KLE, as implemented, among our list of "explicit" estimators of SE, because they do not explicitly estimate the entropy of the "true" distribution over semantic equivalence classes. For instance, Nguyen et al. (2025) describe a formulation of SNNE that equals DSE and Nikitin et al. (2024) show that a kernel exists to recover SE from KLE, but these are not necessarily the configurations used experimentally.

## A.6  BRADLEY-TERRY CONFIDENCE INTERVALS

Gao et al. (2023) offers a procedure for ascertaining CIs about ranks of Bradley-Terry strength estimates, which we cursorily summarize as follows:

Assuming a Bradley-Terry model, let $\beta = \{\beta_1, \ldots, \beta_m\}$ be the true latent strengths of $m$ methods, and let $\widehat{\beta} = \{\widehat{\beta}_1, \ldots, \widehat{\beta}_m\}$ be MLEs of the same. Given a provided level $\alpha$ and a "target" method $i$, we may establish a $(1 - \alpha)$ CI about $\widehat{\beta}_i$, as well as "slightly more conservative" intervals about the remaining strength estimates. Let $n_1$ be the number of resultant intervals whose lower bounds are greater than the upper bound of method $i$'s. Similarly, let $n_2$ be the number of intervals whose upper bounds are less than the lower bound of method $i$'s. The true rank of method $i$'s Bradley-Terry strength is in the integer interval $[n_1 + 1, n - n_2]$ with approximate probability $1 - \alpha$.

Because Proposition 4.1 of Gao et al. (2023) only holds for fixed number of methods/agents/players, we illustrate the "conservative" CIs in Figure 3C. We repeat this procedure for each UQ method to

|  | \multicolumn{4}{c}{$a$} |  |  |  |
| --- | --- | --- | --- | --- |
|  | 0 | 0.01 | 0.1 | 1 |
| $\widehat{|S|}_{Hybrid}$ | [1, 4] | [1, 4] | [1, 4] | [1, 4] |
| NumSets | [7, 11] | [7, 11] | [7, 11] | [7, 11] |
| Good-Turing | [7, 11] | [7, 11] | [7, 11] | [7, 11] |
| $U_{EigV}$ | [1, 3] | [1, 3] | [1, 3] | [1, 3] |
| $\widehat{\mathbb{H}}_{Hybrid}$ | [4, 5] | [4, 5] | [4, 5] | [4, 5] |
| $\widehat{\mathbb{H}}_{Plugin}$ | [7, 11] | [7, 11] | [7, 11] | [7, 11] |
| $\widehat{\mathbb{H}}_{CS}$ | [7, 11] | [7, 11] | [7, 11] | [7, 11] |
| PE | [3, 4] | [3, 4] | [3, 4] | [3, 4] |
| SNNE | [6, 6] | [6, 6] | [6, 6] | [6, 6] |
| KLE | [1, 3] | [1, 3] | [1, 3] | [1, 3] |
| SE | [7, 11] | [7, 11] | [7, 11] | [7, 11] |

Table 3: Employing distinct values of regularization parameter $a$, including zero regularization, we compute MLE Bradley-Terry latent strength scores and establish 95% CIs of each method's latent strength rank via the methods of Gao et al. (2023). Results are identical for all $a$ values considered.

establish CIs for all strength ranks. We refer interested readers to Gao et al. (2023) for further details. The algorithm we use to estimate strength scores employs a regularization parameter, which we take as $a = 0.1$ in the main body of the paper. Table 3, which repeats the analysis of Figure 3C for varying $a$, indicates that the rank order of our Bradley-Terry results is not sensitive to this choice.

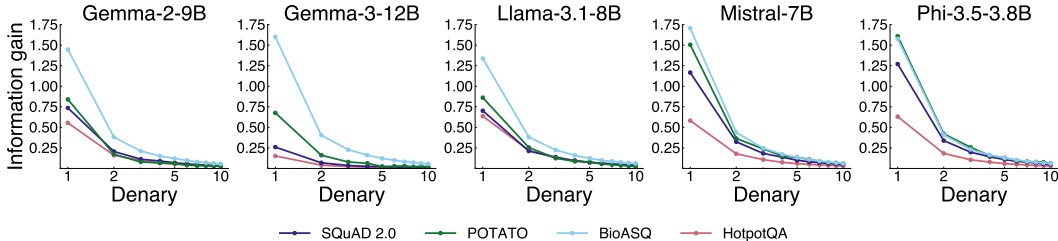

Figure 6: Information gain with respect to the distribution over semantic categories, per denary (group of 10 samples). Because the first denary is novel, its information gain is equal to its plugin entropy. Results are averaged across queries for each dataset. Log scale is used on the x-axis to highlight differences for early denaries.

## B    THE UNDER-SAMPLED REGIME

Consider a simple model of semantic category-abundance, where the probability that a model parameterized by $\theta$ generates a response belonging to the $r^{th}$-most prevalent semantic category $s_r$ in response to a query $q$ follows a Zipfian distribution, similar to some models of species-abundance (Ulrich et al., 2010):

$$p(s_r|q,\theta) = \frac{1}{rH_{|S|}}, \tag{14}$$

where $H_j$ is the $j^{th}$ harmonic number (Kingsley Zipf, 1932). If the expected number of occurrences of $s_r$ in a sample of $n$ responses is less than 1, then $|S| > \frac{n}{H_{|S|}}$. For sample size $n = 10$, we expect at least one semantic category to be unobserved with just $|S| > 4$.

## C    SAMPLE SIZE AND INFORMATION GAIN

Because the true distribution over semantic equivalence classes is not known, we assume that $SE^*$ using $n = 100$ is an adequate representation of the true estimand. In support of this choice, we consider the information gain per denary (i.e., group of ten samples, so the first denary is the first ten samples, etc.) with respect to the distribution over semantic categories - that is, the marginal information about this distribution obtained by observing each successive ten samples (Figure 6). Across models and datasets, this quantity plateaus to low values well before $n = 100$. Predictably, the first denary of samples tends to provide the most information, especially for datasets whose queries can have multiple possible correct semantic categories.

## D    ADDITIONAL UNDERESTIMATION RESULTS

Detailed results for SE underestimation, broken out across four datasets, are shown in Figure 7. Results are qualitatively similar to those in Figure 2, with $\hat{\mathbb{H}}_{Hybrid}$ typically underestimating $SE^*$ less than plugin DSE for practical sample sizes. Additionally, we present a version of Figure 2 that includes white-box SE (Figure 8) and elaborate on the close tracking between SE and plugin DSE in Section E.

$SE^*$ can be exactly zero (e.g., when only one observed semantic category is observed at $n = 100$), hence we exclude such instances from Figures 2, 7, and 8. Occasionally, exceedingly rare semantic categories are revealed early, resulting in coverage-adjusted estimates that substantially exaggerate the distribution's tail mass. We observe a few instances of this phenomenon in our sampling from the Gemma-3-12B model on the POTATO dataset: three queries in POTATO for which Gemma-3-12B elicits a non-zero $SE^*$ (calculated using $n = 100$), but a hybrid DSE (calculated using $n = 10$) that is an order of magnitude larger. Inspection reveals that all three have two semantic categories observed at $n = 100$, and both categories are also observed within the first 10 samples. Of these instances, all have $SE^* < 0.05$ and one has $SE^* < 0.005$. The impacts of this phenomenon are

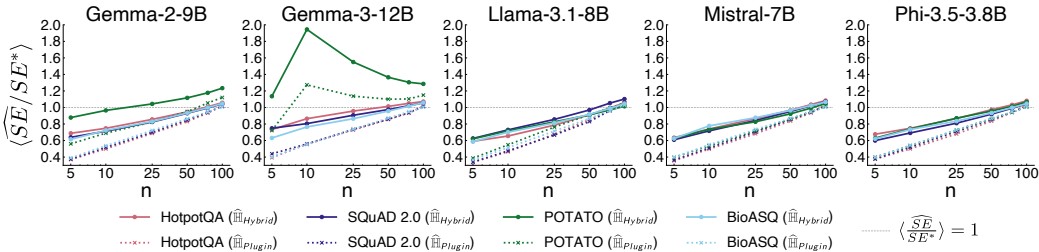

Figure 7: Ratio of semantic entropy (SE) estimator with varying sample size ($n = 5, 10, 25, 50, 75, 100$) to white-box SE with $n = 100$ (denoted $SE^*$), with values below 1 suggesting underestimation (dotted grey line). Estimators shown are the plugin estimator of canonical discrete semantic entropy (DSE; dotted rose, indigo, green, and cyan lines) and the presented DSE estimator of Equation 10 (solid rose, indigo, green, and cyan lines). Results are averaged across queries for each dataset. Log scale is used on the x-axis to highlight differences between estimators with smaller sample sizes. Instances with a denominator of 0 are ignored.

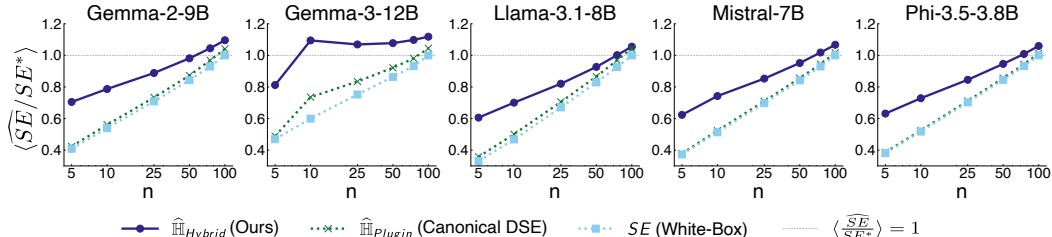

Figure 8: Illustrating underestimation in discrete semantic entropy (DSE) calculation with typical sample sizes. The ratios of semantic entropy (SE) estimators with varying sample size ($n = 5, 10, 25, 50, 75, 100$) to white-box SE with $n = 100$ (denoted $SE^*$) are shown, with values below 1 suggesting underestimation (dotted grey line). The estimators displayed are the plugin estimator of canonical DSE (i.e., Equation 5, dotted green line), the "hybrid" SE estimator of Equation 10 (solid indigo line), and white-box SE (Equation 3, dotted cyan line). Results are averaged across queries within each dataset, then uniformly averaged across datasets. Log scale is used on the x-axis to highlight differences between estimators with smaller sample sizes. Instances with a denominator of 0 are ignored.

visible in Figures 2 and 7, where the curve for Gemma-3-12B on POTATO deviates from the pattern observed for other model-dataset pairs. If we exclude the most extreme instance by visualizing only instances with $SE^* \geq 0.005$, all model-dataset pairs exhibit the familiar behavior (Figure 9).

## E  HOW DISTINCT ARE CANONICAL DSE AND SE?

In Figures 8 and 9, we observe that SE underestimates $SE^*$, which is unsurprising, as it also invokes the negatively biased plugin entropy estimator (Equation 3). It is surprising, however, that SE appears to track plugin DSE so closely, since SE's probabilities are calculated via RBMCI (Equation 4), versus plugin DSE's empirical class frequencies. In Figure 3, SE does not appear definitively superior to plugin DSE overall on the downstream incorrectness detection task. Similarly, Farquhar et al. (2024) and Nikitin et al. (2024) report little difference in incorrectness detection performance metrics (e.g., AUROC) between SE and DSE.

In Table 4, we compare the empirical distribution of semantic class frequencies to the one induced by RBMCI, using the data from the experiment supporting Figure 3. Our metrics of comparison are their Kullback-Leibler divergence (KL) (Kullback

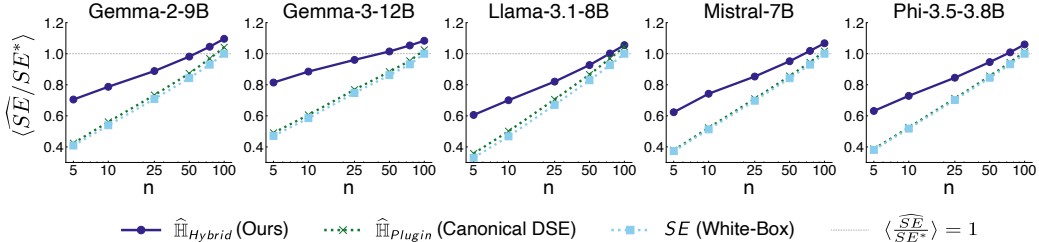

Figure 9: Illustrating underestimation in discrete semantic entropy (DSE) calculation with typical sample sizes. The ratios of semantic entropy (SE) estimators with varying sample size ($n = 5, 10, 25, 50, 75, 100$) to white-box SE with $n = 100$ (denoted $SE^*$) are shown, with values below 1 suggesting underestimation (dotted grey line). The estimators displayed are the plugin estimator of canonical DSE (i.e., Equation 5, dotted green line), the "hybrid" SE estimator of Equation 10 (solid indigo line), and white-box SE (Equation 3, dotted cyan line). Results are averaged across queries within each dataset, then uniformly averaged across datasets. Log scale is used on the x-axis to highlight differences between estimators with smaller sample sizes. Instances with exceedingly small denominators (i.e., $SE^* < 0.005$) are ignored.

& Leibler, 1951), total variation distance (TVD) (Jacobs, 2018), and the mean absolute error (MAE), all computed by comparing each method's class-wise probability vectors. All three are exceedingly small across models and datasets; for instance, the KL divergence is nearly always well below 0.01 (Table 4). We consider this finding in light of Figure 8, where the curves for DSE-based methods are visually impacted by an anomalous sampling result, but that of SE is not. We conclude that, while SE and plugin DSE may compute plugin entropy over nearly identical distributions, on average, sequence log-probabilities can help RBMCI down-weight scenarios when the sample over-represents exceptionally rare events.

## F ADDITIONAL INCORRECTNESS DETECTION RESULTS

Detailed results for incorrectness classification across three datasets, five LLMs, and eleven UQ methods are shown in Figure 10. Error bars are 95% CIs about empirical AU-ROC values, calculated via DeLong's method (DeLong et al., 1988; Sun & Xu, 2014).

| Model | Dataset | KL | TVD | MAE |
|---|---|---|---|---|
| Gemma-2-9B | HotpotQA | 0.001 | 0.006 | 0.005 |
| | SQuAD 2.0 | 0.002 | 0.009 | 0.006 |
| | POTATO | 0.004 | 0.021 | 0.014 |
| | BioASQ | 0.004 | 0.021 | 0.012 |
| Gemma-3-12B | HotpotQA | 0.000 | 0.001 | 0.001 |
| | SQuAD 2.0 | 0.000 | 0.002 | 0.002 |
| | POTATO | 0.001 | 0.004 | 0.004 |
| | BioASQ | 0.001 | 0.011 | 0.006 |
| Llama-3.1-8B | HotpotQA | 0.009 | 0.025 | 0.018 |
| | SQuAD 2.0 | 0.010 | 0.029 | 0.019 |
| | POTATO | 0.008 | 0.034 | 0.021 |
| | BioASQ | 0.018 | 0.053 | 0.025 |
| Mistral-v0.3-7B | HotpotQA | 0.001 | 0.009 | 0.007 |
| | SQuAD 2.0 | 0.002 | 0.015 | 0.010 |
| | POTATO | 0.004 | 0.023 | 0.016 |
| | BioASQ | 0.004 | 0.027 | 0.014 |
| Phi-3.5-3.8B | HotpotQA | 0.003 | 0.013 | 0.010 |
| | SQuAD 2.0 | 0.002 | 0.013 | 0.010 |
| | POTATO | 0.012 | 0.039 | 0.028 |
| | BioASQ | 0.005 | 0.029 | 0.016 |

Table 4: Comparing semantic equivalence class probability distributions, induced by empirical frequencies (as in black-box DSE) or via Equation 4 (as in white-box SE). The metrics considered are Kullback-Leibler divergence (KL), total variation distance (TVD), and the mean absolute error between the class-wise probability vectors (MAE). For each model, we average results over the questions of: HotpotQA, the answerable questions of SQuAD 2.0, POTATO, and BioASQ. Values are rounded to four decimal places to adequately represent the smallest entries.

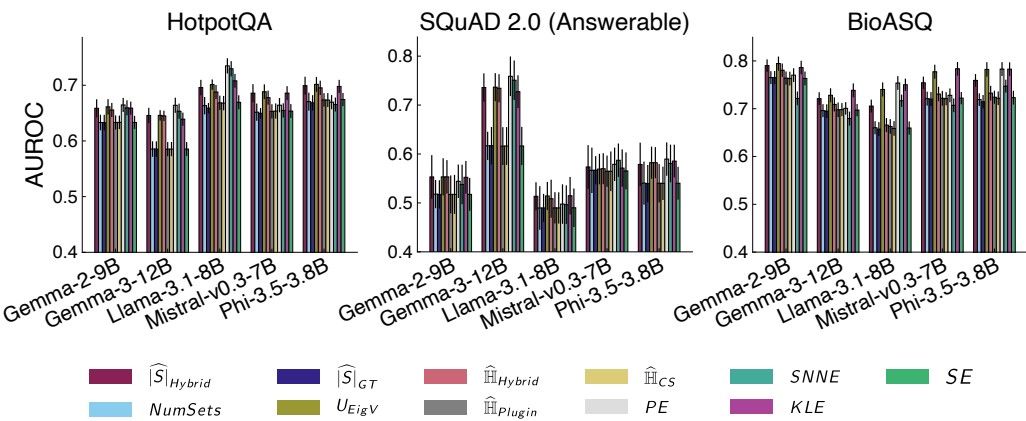

Figure 10: Performance of incorrectness classification on single-answer QA for uncertainty measures using $n = 10$ samples. Methods considered include black-box semantic alphabet size estimators ($\widehat{|S|}_{Hybrid}$, NumSets, $\widehat{|S|}_{GT}$, $U_{EigV}$), explicit discrete semantic entropy (DSE) estimators ($\widehat{\mathbb{H}}_{Hybrid}$, $\widehat{\mathbb{H}}_{Plugin}$, $\widehat{\mathbb{H}}_{CS}$), and other uncertainty estimators, which include white-box (PE, SE) and black-box methods (SNNE, KLE). Error bars reflect 95% CIs about AUROC, via DeLong's method.

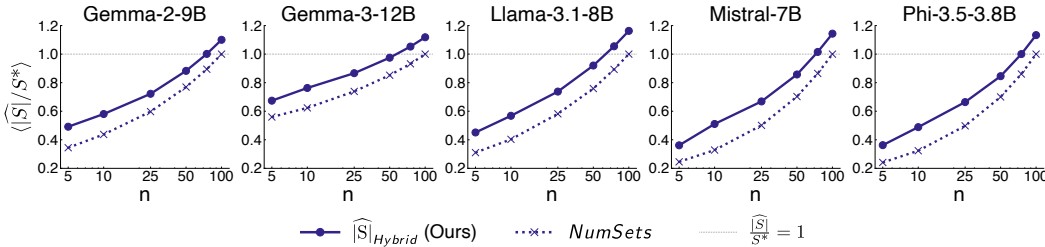

Figure 11: Illustrating underestimation in alphabet size estimation with typical sample sizes. The ratios of alphabet size estimators with varying sample size ($n = 5, 10, 25, 50, 75, 100$) to the observed number of semantic categories with $n = 100$ (denoted $S^*$) are shown, with values below 1 suggesting underestimation (dotted grey line). The estimators displayed are NumSets, used by the plugin estimator of canonical discrete semantic entropy (dotted indigo line) and the "hybrid" alphabet size estimator of Equation 9 (solid indigo line). Results are averaged across queries within each dataset, then uniformly averaged across datasets. Log scale is used on the x-axis to highlight differences between estimators with smaller sample sizes.

## G    UNDERESTIMATION OF
SEMANTIC ALPHABET SIZE

In Figure 2 we exhibit underestimation of SE, using the ratio of DSE estimators (using varying sample sizes) to $SE^*$. We provide a similar illustration in Figure 11: NumSets, the implicit alphabet size estimator of canonical DSE, underestimates the number of semantic categories for typical sample sizes, and the the hybrid semantic alphabet size estimator does so by less.

## H    IMPLEMENTATION VARIATIONS

The relevant LLM UQ literature is not uniform in experimental setup. For instance, the plurality of considered works use $n = 10$ sampled responses per query (Kuhn et al., 2023; Farquhar et al., 2024; Nikitin et al., 2024; Nguyen et al., 2025), but Lin et al. (2024) use $n = 20$. Some works vary prompting templates across datasets (e.g., zero-shot vs. multi-shot prompting) (Kuhn et al.,

2023; Lin et al., 2024). A variety of methods for automated LLM incorrectness evaluation have been employed, including thresholded ROUGE-L (Kuhn et al., 2023), binary LLM-as-judge (Farquhar et al., 2024), thresholded consistency-based LLM-as-judge Lin et al. (2024), and BERTScore (Nguyen et al., 2025). Although it is out of this work's scope to exhaustively ablate all such choices, we attempt to be explicit in describing our selections to support future work.

## I LLM USAGE

During this project, LLMs were employed at times for coding assistance.

