# OpenReview forum: "Estimating Semantic Alphabet Size for LLM Uncertainty Quantification"
_ICLR.cc/2026/Conference — ICLR 2026 Poster_

### Official Review · Reviewer_RUVy · 2025-10-22

**Soundness:** 4
**Presentation:** 4
**Contribution:** 3
**Rating:** 8
**Confidence:** 5

**Summary:**

I really like this paper. It takes a fairly simple premise: sampling-based estimators of the clustering distribution will tend to miss the tail, leading to systemic underestimation of semantic entropy. But: We have robust estimators for the size of the tail. So, we can get modified estimators of either the number of semantic clusters, or the semantic entropy. And, these modified estimators lead to improved performance.

**Strengths:**

* It’s a simple idea... in a good way. It identifies a clear problem, finds parallels in the species sampling literature for heavy tailed distributions, and proposes a simple, interpretable solution.
* It leads to improvement.
* The evaluation is very thoughtful (one of the best discussions of how to evaluate UQ methods I have seen)
* The paper is clear and well-written

**Weaknesses:**

* It’s not strictly improving on SoTA... but it is compatible with SoTA using a much simpler and more interpretable paradigm
* The idea of using a Good-Turing estimator has been explored in related problems in the literature, as acknowledged by the authors.

**Questions:**

* It’s interesting that you find that the adjusted number of sets outperforms the adjusted semantic entropy. It would be nice to see how the “ground truth” number of sets compares with “ground truth” semantic entropy... if in this setting ground truth number of sets outperforms ground truth semantic entropy, it might suggest something interesting about miscalibration of the LLM.
* I’d like to see (maybe in the discussion) discussion of how one might extend this to white-box SE estimation (which will tend to suffer from the same problem, albeit to a lesser extent).

---

> ### Author Response · Authors · 2025-11-25
> **Author Response to Reviewer RUVy**
>
> We are thrilled that the Reviewer **“really like(s) this paper,”** and finds it **“clear and well-written,”** with a **“thoughtful”** evaluation that includes **“one of the best discussions of how to evaluate UQ methods [the Reviewer has] seen.”** This is high praise that the authors graciously appreciate. We respond to the Reviewer’s feedback below:
>
> > <<Question 1>>: on evaluating “”ground truth” number of sets and SE for incorrectness detection, and potential implications regarding LLM calibration
>
> Our incorrectness detection results use n=10 responses per query, whereas “ground truth” semantic entropy and “ground truth” number of sets require drawing 100 samples per query. Unfortunately, this is intractable in the time remaining. The topic of LLM miscalibration is intriguing, however, and mentioned by Nikitin et al., who report a 70% AUROC win rate for DSE vs. SE (their Figure 3a), which declines substantially when only considering non-instruction-tuned models (Appendix Figure D.5a,b). They also observe that KLE “only marginally outperforms when evaluated on non-instruction-tuned models” and suggest that this may be due to miscalibration of instruction-tuned models, as you allude [1].
>
> In responding to this comment, we discovered a coding error that resulted in our black-box entropy estimators using Rao-Blackwellized class probabilities, rather than empirical frequencies. **Fortunately, this did not impact our findings at all**, as illustrated in the updated paper. This itself is curious! We investigate further in Appendix Section E, contextualizing this with other findings, and conclude that **“while SE and plugin DSE may compute plugin entropy over nearly identical distributions, on average, sequence log-probabilities can help RBMCI down-weight scenarios when the sample over-represents exceptionally rare events.”** This may explain why prior work often finds SE and DSE performing quite similarly on downstream tasks (e.g., Figure 2 and Supplementary Figure 5 of Farquhar et al. [2], Tables 2 and 3 of Nguyen et al. [3], Table 1 of Nikitin et al. [1]).
>
> > <<Question 2>>: on extensions to white-box SE estimation
>
> This is an interesting point, which we have briefly incorporated into the Discussion.
>
> > <<Weakness 1>>: “It’s not strictly improving on SoTA... but it is compatible with SoTA using a much simpler and more interpretable paradigm”
>
> While this is true, we disagree that it is a weakness, because we do not necessarily set out to establish a new SOTA for hallucination detection. First, **we illustrate a property of a widely-referenced UQ method** (plugin DSE’s underestimation) and **demonstrate improvement by adjusting for semantic alphabet size**. Our subsequent experiment confirms that this approach **also leads to stronger incorrectness detection**, compared to other explicit semantic entropy estimators. Our observation that semantic alphabet size estimation alone can outperform many other approaches is surprising and interesting, but was not our original goal.
>
> > <<Weakness 2>>: on the GT estimator having been explored in related problems in the literature
>
> This is also true, but to the best of our knowledge, the application is novel in the LLM UQ setting.
>
>
> We found the Reviewer’s feedback thoughtful and highly relevant and feel that the included recommendations have led to an even stronger paper. Thank you for your time and consideration.
>
>
> **References**
>
> [1] Nikitin, Alexander, Jannik Kossen, Yarin Gal, and Pekka Marttinen. "Kernel Language Entropy: Fine-grained Uncertainty Quantification for LLMs from Semantic Similarities." Advances in Neural Information Processing Systems 37 (2024): 8901-8929.
>
> [2] Farquhar, Sebastian, Jannik Kossen, Lorenz Kuhn, and Yarin Gal. "Detecting hallucinations in large language models using semantic entropy." Nature 630, no. 8017 (2024): 625-630.
>
> [3] Nguyen, Dang, Ali Payani, and Baharan Mirzasoleiman. "Beyond Semantic Entropy: Boosting LLM Uncertainty Quantification with Pairwise Semantic Similarity." Findings of the Association for Computational Linguistics (2025): 4530–4540.

---

### Official Review · Reviewer_SfCH · 2025-10-31

**Soundness:** 3
**Presentation:** 3
**Contribution:** 2
**Rating:** 6
**Confidence:** 5

**Summary:**

Goal:
- This paper addresses a specific issue in discrete semantic entropy (DSE), a widely used black-box UQ method.
- The authors show that DSE tends to underestimate uncertainty when the response sample size is small, because it often fails to capture all unseen semantic clusters, therefore underestimating the number of distinct meaning clusters in its outputs (**semantic alphabet size**).

Empirical Validation:
- To verify this bias, the authors compare DSE values from small-sample settings against 100-sample semantic entropy (treated as ground-truth uncertainty).
- Across all model–dataset pairs, DSE consistently produces lower uncertainty estimates as sample size decreases.

Method:
To better estimate the semantic alphabet size, the paper considers 2 existing approaches:
1. A Good–Turing coverage estimator, adapted from population ecology, which corrects for unseen categories in finite samples
2. A spectral method based on eigenvalue decomposition (Lin et al.), which estimates alphabet size from the number of non-zero eigenvalues of a semantic similarity matrix.
The authors then propose a hybrid estimator that combines these two strategies:
- when Good–Turing coverage becomes unreliable (e.g., each cluster appears only once), the method defaults to the spectral estimate;
- otherwise, it uses a weighted combination to mitigate underestimation bias.
Using this hybrid alphabet size, they further define a hybrid semantic entropy estimator, extending classic ecological entropy formulations to the semantic domain.

Evaluation:
1. Following standard UQ evaluation protocols, the study benchmarks ten uncertainty estimators across four LLM families and four datasets.
2. Because raw AUROC scores can vary with model and dataset, the authors adopt a Bradley–Terry latent strength model to aggregate pairwise AUROC comparisons into an overall ranking with confidence intervals—an extension of the win-rate framework proposed by Nikitin et al. (2024).
Overall, results show that the hybrid semantic alphabet size estimator performs most consistently across settings, while the hybrid semantic entropy estimator reduces but does not fully eliminate the negative bias in DSE.

**Strengths:**

1. The problem itself is important.
2. The solution is simple.
3. The evaluation considers the weakness of AUROC and proposes the Bradley-Terry latent strength scores, which make the evaluation more reliable.
4. The paper is easy to understand and well-structured. I like its simplicity and enjoy reading it.

**Weaknesses:**

1. Limited methodological novelty and analysis.
- The proposed hybrid estimator mainly combines two existing techniques
   -  the Good–Turing coverage estimator from population ecology
   - a spectral eigenvalue-based method
I am not saying that combining prior ideas is not good, and my point is that it can be done with a deeper analysis of each individual methods and the hybrid operation can be done more adaptively.
- For example, the Good–Turing approach is well established for ecological sampling, yet the sampling distribution of LLM-generated responses can be very different. It remains unclear why this method should perform reliably in the LLM setting or under what conditions it might fail.
- Similarly, the hybrid design could be more principled — for instance, by characterizing when each component estimator is more accurate, rather than simply prioritizing the larger estimate under certain cases. This max operation inherently assumes that both the spectral-based method and Good-Turing coverage-based methods are also underestimating the ground truth semantic cluster size. But is this assumption true?

2. Inconsistent motivation and results.
The empirical validation (Figure 2 and table 1) assumes that semantic entropy computed with 100 samples represents the ground truth. However, this assumption itself is not rigorously justified. Moreover, the paper’s motivation is to improve DSE-based uncertainty estimation, yet the final results show that DSE-derived methods (including the hybrid entropy estimator) still lag behind KLE and the semantic alphabet size estimator in overall performance (always ranking from 7-10). This weakens the narrative that improving DSE necessarily leads to better uncertainty quantification.

**Questions:**

1. Equation (8) appears to be missing the definition or notation for $p_i$.
2. I have multiple questions regarding the experiment results (Figure 3):
- Why was white-box semantic entropy not included in the comparison, since you also consider the PE?
- In addition, both hybrid semantic entropy ($\hat{H}_\text{hybrid}$) and discrete semantic entropy (DSE) show notably lower rankings (around 7–10). Could you elaborate on why these methods perform substantially worse than others? Conceptually, both $\hat{H}_\text{hybrid}$ and DSE should still be affected by the same underestimation bias that motivates your paper. If the semantic alphabet size were estimated more accurately, shouldn’t this also mitigate the underestimation problem in DSE to some extent?

---

> ### Author Response · Authors · 2025-11-25
> **Author Response to Reviewer SfCH (Part 1/2)**
>
> We are delighted to hear that the Reviewer considers our problem of interest **“important,”** found the paper **“easy to understand and well-structured,”** and **“like[ed] its simplicity and enjoy[ed] reading it.”** Similarly, we are pleased that the Reviewer noted our refinement of LLM UQ evaluation methods as a strength. We are proud of these aspects and grateful that the Reviewer found them worth highlighting.
>
> The Reviewer poses three questions, which reflect careful reading and have helped us improve the presentation of our work:
>
> > <<Question 1>>: Equation (8) appears to be missing the definition or notation for p_i.
>
> Thank you for bringing our attention to this oversight. We have updated the paper with the definition of $\hat{p}_i$.
>
> > <<Question 2>>: on SE not being included the comparison
>
> We have added white-box SE to the comparison. Our hybrid entropy estimator achieves a higher uncertainty-aware rank than white-box SE, as do several other methods considered.
>
> Perhaps more interestingly, SE and plugin DSE tend to track each other very closely. We examine this in Appendix Section E, where we conclude that “while SE and plugin DSE may compute plugin entropy over nearly identical distributions, on average, sequence log-probabilities can help RBMCI down-weight scenarios when the sample over-represents exceptionally rare events.” This may explain why prior work often finds SE and DSE performing quite similarly on downstream tasks (e.g., Figure 2 and Supplementary Figure 5 of Farquhar et al. [1], Tables 2 and 3 of Nguyen et al. [2], Table 1 of Nikitin et al. [3]).
>
> > <<Question 3>>: on both hybrid and plugin DSE sharing the lowest rank CI
>
> We wish to clarify that hybrid and plugin DSE do not share the lowest rank band. Our hybrid DSE estimator **substantially reduces the reported underestimation** for practical sample sizes (Figure 2), resulting in **more accurate entropy estimation** (Table 1), and better incorrectness detection compared to plugin DSE (Figure 3). We hope this answers the Reviewer’s question. Please let us know if we have misunderstood the Reviewer’s intent.
>
>
> Finally, we would like to respond to a few other points raised by the Reviewer:
>
> > <<Weakness 1>>: on limited novelty
>
> We agree that the proposed hybrid estimators integrate several existing techniques, and we believe that this synthesis from multiple disciplines is itself a valuable contribution for LLM UQ. We would like to emphasize, however, that **our work includes several other contributions,** including an **empirical demonstration of plugin DSE’s underestimation** (which alone may impact works that rely on DSE) and a **refinement of UQ evaluation**, (which Reviewer `RUVy` referred to as “one of the best discussions of how to evaluate UQ methods [they] have seen”).
>
> > <<Weakness 2, part 1>>: on the use of n=100 for SE^*
>
> The n=100 we currently use is already an order of magnitude larger than found in prior related works we reviewed, which almost exclusively use n=10 (Section 4.1 and Appendix Section G). Scaling even larger is computationally infeasible: in addition to the cost LLM sampling, the semantic clustering algorithm has combinatorial complexity with respect to the cost of entailment classification.
>
> To further address this comment, we compute information gain per denary (i.e., the amount of additional information the last 10 samples provide us about the distribution over semantic categories). Across models and datasets, this becomes quite small well before n=100 (Figure 6).
>
> [Response continued in next comment]

---

> ### Author Response · Authors · 2025-11-25
> **Author Response to Reviewer SfCH (Part 2/2)**
>
> > <<Weakness 2, part 2>>: “the final results show that DSE-derived methods (including the hybrid entropy estimator) still lag behind KLE and the semantic alphabet size estimator in overall performance (always ranking from 7-10)”
>
> Our hybrid DSE estimator does not share the low-end rank band with the other DSE-derived methods; instead, it occupies the [4, 5] rank band. This illustrates that **our method substantially improves DSE estimators for incorrectness detection**. It is true that KLE and two alphabet size estimators outperform the hybrid DSE estimator overall at incorrectness detection, but we do not see this as a weakness - that alphabet size estimators perform as well as they do is a surprising and interesting finding.
>
>
> We are grateful to the Reviewer for lending their expertise and hope that our responses have satisfied the Reviewer’s remaining questions.
>
> **References**
>
> [1] Farquhar, Sebastian, Jannik Kossen, Lorenz Kuhn, and Yarin Gal. "Detecting hallucinations in large language models using semantic entropy." Nature 630, no. 8017 (2024): 625-630.
>
> [2] Nguyen, Dang, Ali Payani, and Baharan Mirzasoleiman. "Beyond Semantic Entropy: Boosting LLM Uncertainty Quantification with Pairwise Semantic Similarity." Findings of the Association for Computational Linguistics (2025): 4530–4540.
>
> [3] Nikitin, Alexander, Jannik Kossen, Yarin Gal, and Pekka Marttinen. "Kernel Language Entropy: Fine-grained Uncertainty Quantification for LLMs from Semantic Similarities." Advances in Neural Information Processing Systems 37 (2024): 8901-8929.

---

### Official Review · Reviewer_foRW · 2025-10-31

**Soundness:** 3
**Presentation:** 3
**Contribution:** 3
**Rating:** 2
**Confidence:** 3

**Summary:**

This paper addresses the problem of underestimating semantic entropy (SE) as an uncertainty estimate for black-box LLMs. The authors propose an interpretable semantic alphabet size estimator which, when used to adjust the discrete SE estimator for sample coverage, yields more accurate uncertainty estimation.

**Strengths:**

- The proposed semantic alphabet size estimator effectively mitigates the underestimation bias of discrete semantic entropy (DSE) in the few-sample regime.
- The method is simple and interpretable, avoiding the complexity and hyperparameter dependence of recent SE extensions.
- Empirically, it improves UQ performance and matches or exceeds the performance of SOTA approaches in the back-box setting.

**Weaknesses:**

- My main concern is that the proposed method is largely an adaptation of existing estimators to the SE setting. While the idea is insightful, it constitutes an incremental improvement over Farquhar et al. (2024) rather than a fundamentally new theoretical contribution.
- The paper treats semantic clusters as fixed and does not analyze sensitivity to the clustering procedure (e.g., entailment thresholds or NLI model biases). Since clustering quality directly affects entropy estimates, this omission is a potential confounder that limits the scope of the contribution.
- Experiments are restricted to relatively small models (<10B parameters), which raises questions about generalization to larger, frontier LLMs. Moreover, the paper does not compare against computationally cheaper uncertainty measures that avoid expensive sampling (e.g., G-NLL [1]) and which outperform SE and DSE with a single sample.
Minor point: The paper inconsistently uses the abbreviations (like SE and DSE), alternating between the abbreviated and full forms without clear rationale, which makes it harder to follow.

---

[1] Lukas Aichberger, Kajetan Schweighofer, and Sepp Hochreiter. Rethinking uncertainty estimation in natural language generation. arXiv preprint arXiv:2412.15176, 2024.

[2] Sebastian Farquhar, Jannik Kossen, Lorenz Kuhn, and Yarin Gal. Detecting hallucinations in large language models using semantic entropy. Nature, 2024.

**Questions:**

- How quickly does the “white-box” SE estimate converge to the "true" SE as the number of samples increases? Including this in Figure 2 would be interesting.
- Are 100 samples sufficient to approximate the true distribution over semantic clusters? An ablation with larger sample sizes (e.g., 1 k) could strengthen the claim that this is a valid reference.
- How sensitive are the results to the semantic clustering method (e.g., choice of NLI model, thresholding, or entailment aggregation)?
- The hybrid estimator uses a max(·) rule combining two estimators. Why is this choice theoretically justified versus, say, a weighted or Bayesian combination?

---

> ### Author Response · Authors · 2025-11-25
> **Author Response to Reviewer foRW (Part 1/2)**
>
> We would like to express sincere gratitude to Reviewer foRW for providing their time and insight to the review process. We are encouraged that the Reviewer finds our approach
> **“effectively mitigates the underestimation bias of discrete semantic entropy (DSE) in the few-sample regime,”** does so in a **“simple and interpretable”** manner, and **rates the soundness, presentation, and contribution of our work as “good.”** We respond to the Reviewer’s questions below:
>
> > <<Question 1>>: on the convergence of “white-box” SE to the "true" SE
>
> Thank you for this suggestion; we have taken your advice and run this experiment (Figure 9). The behavior is generally similar to that of plugin DSE. This observation has been included in a larger discussion about the surprising similarity between plugin DSE and SE (Appendix Sections D and E).
>
> > <<Question 2>>: on whether n=100 is sufficient, and suggesting an ablation with n=1000
>
> The n=100 we currently use is already an order of magnitude larger than found in prior related works we reviewed, which almost exclusively use n=10 (Section 4.1 and Appendix Section G). Scaling even larger is computationally infeasible: in addition to the cost LLM sampling, the semantic clustering algorithm has combinatorial complexity with respect to the cost of entailment classification.
>
> To further address this comment, we compute information gain per denary (i.e., the amount of additional information the last 10 samples provide us about the distribution over semantic categories). Across models and datasets, this becomes quite small well before n=100, suggesting that n=1000 is unlikely to yield a more cost-effective reference (Figure 6).
>
> > <<Question 3>>: on the sensitivity of results to semantic clustering method, and
> > <<Weakness 2>>: suggesting analyzing sensitivity to entailment thresholds or NLI model biases
>
> Semantic clustering is not the subject of our study, and we inherit our clustering algorithm from prior work. That said, we agree that clustering quality could affect entropy estimates; in Table 2, **we already compare bidirectional entailment clustering methods against human-annotated ground truth,** finding that the one we use is comparable to a more costly LLM-based alternative. This is similar to the comparison performed in the Supplementary Materials of Farquhar et al. [1]. It is not clear to us what analyzing sensitivity to “NLI model biases” would entail in this context. Nonetheless, we have **extended this analysis to include five baselines,** which generally perform worse.
>
> > <<Question 4>>: on justification of the max(·) rule, and whether a weighted or Bayesian combination would be more appropriate
>
> Our hybrid estimator encodes logic about the scenarios in which each estimator is preferable. For instance, the Good-Turing estimator is undefined when all samples belong to distinct semantic equivalence classes, and the spectral estimator can be below a lower bound for |S|.
>
> Upon recommendation by Reviewer T6sp, we have added a Figure 2-like plot for semantic alphabet size estimators to the Appendix (Figure 11). Because the hybrid alphabet size estimator still underestimates the true alphabet size for typical sample sizes (but less so than the implicit alphabet size estimator of canonical DSE), we believe that replacement of the max(·) operator with a weighted combination could underestimate the true alphabet size by more than ours, in the setting of interest (since max(A, B) \geq xA + yB \forall x+y=1).
>
> > <<Weakness 1>>: on the estimators constituting incremental improvement
>
> While the proposed hybrid estimators indeed integrate several existing techniques, we believe this synthesis from multiple disciplines is itself a valuable contribution. We also emphasize that **our work includes several other contributions,** including an **empirical demonstration of plugin DSE’s underestimation** (which alone may impact works that rely on plugin DSE) and a **refinement of UQ evaluation**, (which Reviewer `RUVy` referred to as “one of the best discussions of how to evaluate UQ methods [they] have seen”).
>
> > <<Weakness 3, part 1>>: experiments being conducted on models with <10B parameters
>
> To address this concern, **we have run additional experiments to include a >10B parameter model** (Gemma-3-12B). In the updated Table 1, for example, **our hybrid estimator still provides the most accurate semantic entropy estimate.** We do observe differences in the “entropy ratio” curves (Figure 2) for Gemma-3-12B, but this appears to be largely driven by a single highly anomalous result in the POTATO dataset, which we address in Appendix Section D. We hope this increased breadth helps reinforce the reliability of our results.
>
> [Response continued in next comment]

---

> ### Author Response · Authors · 2025-11-25
> **Author Response to Reviewer foRW (Part 2/2)**
>
> > <<Weakness 3, part 2>>: “Moreover, the paper does not compare against computationally cheaper uncertainty measures that avoid expensive sampling (e.g., G-NLL [1]) and which outperform SE and DSE with a single sample.”
>
> Our paper does compare against a computationally cheaper uncertainty measure, namely predictive entropy (PE), a white-box method that outperforms SE and plugin DSE overall in our experiments. Although it does not avoid multiple sampling, it does avoid the computationally expensive clustering step. Moreover, our work focuses on black-box methods, which G-NLL is not. It is also worth noting that G-NLL requires us to greedily sample from all queries/datasets/models, so it cannot be easily incorporated in the time remaining. Future work may focus on other white-box methods, such as G-NLL.
>
> > <<Weakness 3, part 3>>: on alternation between abbreviated and full forms of of acronyms
>
> We appreciate the Reviewer’s careful eye and have revised the paper to more consistently use abbreviations.
>
>
> We reiterate our gratitude to the Reviewer for their time and consideration. We are confident that this dialogue has resulted in a stronger presentation of our work.
>
> **References**
>
> [1] Farquhar, Sebastian, Jannik Kossen, Lorenz Kuhn, and Yarin Gal. "Detecting hallucinations in large language models using semantic entropy." Nature 630, no. 8017 (2024): 625-630.
>
> [2] Nguyen, Dang, Ali Payani, and Baharan Mirzasoleiman. "Beyond Semantic Entropy: Boosting LLM Uncertainty Quantification with Pairwise Semantic Similarity." Findings of the Association for Computational Linguistics (2025): 4530–4540.
>
> [3] Nikitin, Alexander, Jannik Kossen, Yarin Gal, and Pekka Marttinen. "Kernel Language Entropy: Fine-grained Uncertainty Quantification for LLMs from Semantic Similarities." Advances in Neural Information Processing Systems 37 (2024): 8901-8929.

---

### Official Review · Reviewer_T6sp · 2025-11-07

**Soundness:** 3
**Presentation:** 3
**Contribution:** 2
**Rating:** 6
**Confidence:** 3

**Summary:**

Authors propose a better (less biased) estimator for semantic-alphabet-size in the semantic-entropy-related methods for UQ estimation in NLG tasks.
They show empirically that their semantic-alphabet-size is competitive with some of the SOTA methods.

**Strengths:**

S1. The contribution is principled and to my best judgement technically correct.

S2. The empirical performance of the proposed method is strong.

S3. I find the empirical evaluation methodology sound.

**Weaknesses:**

W1. The evaluation could use more datasets (despite being sound methodologically).
There are codebases available which would make the process rather straightforward, given the method proposed is rather simple to implement once the Semantic-Clustering is computed: https://github.com/AlexanderVNikitin/kernel-language-entropy. + maybe SimpleQA?(https://openai.com/index/introducing-simpleqa/)

W2. Not sure how "significant" the contribution will prove to be, but I hate to judge this criterium, so please treat this complaint as secondary. If this was judged by TMLR criteria, my overall score would be an "accept".

**Questions:**

Q1. Would it possible to create a Figure-2-like plot but for semantic-alphabet-size estimators?

Q2. Nitpick: could you please change the font in Fig 1 to a Serif font?

---

> ### Author Response · Authors · 2025-11-25
> **Author Response to Reviewer T6sp**
>
> We thank Reviewer T6sp for their careful examination and constructive feedback. We are thrilled that the Reviewer finds our work **“principled,”** exhibiting **“strong” performance**, and evaluating in a manner that is **“sound methodologically.”** The Reviewer raises two questions and two potential weaknesses, which we address below:
>
> > <<Question 1>>: requesting a Figure-2-like plot for alphabet size estimators
>
> We have added such a figure to the Appendix (Figure 11). Therein, we observe a similar pattern: on average, the hybrid alphabet size estimator underestimates the number of semantic categories by less than the implicit alphabet size estimator of canonical DSE (NumSets).
>
> > <<Question 2>>: requesting a serif font for Figure 1
>
> We have updated Figure 1 to use a serif font.
>
> > <<Weakness 1>>: on using more datasets
>
> We consider our dataset selection fairly broad, deliberately including ones with queries that have multiple possible correct answers. Unfortunately, increasing the experimental scale further in the remaining time is difficult, due to the large number of samples (n=100, an order of magnitude larger than “typical” in prior work, to support Figure 2 and Table 1), combinatorial complexity of semantic clustering, and computational constraints. Nevertheless, we have run additional experiments to include a >10B parameter model (Gemma-3-12B). We hope this increased breadth helps reinforce our results.
>
> > <<Weakness 2>>: on the significance of the contribution
>
> We are pleased to hear that the Reviewer would recommend acceptance by TMLR criteria. Recognizing that forecasting impact is an innately challenging task, we would like to briefly highlight a few aspects that we feel contribute to the significance of our work:
> - We present, to our knowledge, **the first empirical demonstration of plugin DSE’s underestimation**, which alone may impact works that rely on DSE;
> - We propose **new estimators** to address the above;
> - We offer a **refinement of UQ evaluation**, which Reviewer `RUVy` referred to as “one of the best discussions of how to evaluate UQ methods [they] have seen”; and
> - Our updated paper includes a discussion of the surprising similarity between SE and DSE (Appendix Section E), which may be of independent interest.
>
> Please let us know if other questions or suggestions remain.

---

### Author Response · Authors · 2025-11-25
**Overall Author Comment**

We are grateful to the Reviewers for their time spent considering our work, and thrilled to have received encouraging comments. Among them, Reviewer `T6sp` described our work as **“principled,”** and Reviewer `foRW` found that our approach addressed plugin DSE’s underestimation **“effectively”** in a **“simple and interpretable”** manner. Several Reviewers highlighted our evaluation approach as **“sound methodologically”** (`T6sp`),  **“more reliable”** (`SfCH`), **“thoughtful”** (`RUVy`), and including **“one of the best discussions of how to evaluate UQ methods I have seen”** (`RUVy`). Finally, Reviewers `SfCH` and `RUVy` referred to the writing itself, finding our paper **“easy to understand and well-structured”** and **“clear and well-written.”**

We also received constructive questions and comments and thank the Reviewers for their patience, as we prepared a thorough response to their feedback and our own inquiries. Our updated submission includes **increased experimental scale** to include a 12B-parameter model (Gemma-3-12B), **five extra baselines** (HDBSCAN and four DBSCAN variations) in the semantic clustering evaluation, **four additional figures** (Figures 6, 8, 9, 11), **supplemental analysis** (Appendix Section E) examining the practical distinctions between SE and plugin DSE, and various textual improvements. After repeating experiments with the larger model and correcting a plotting error, our core findings are:
- **Plugin DSE underestimates the “true” semantic entropy**, on average, for typical sample sizes (Figure 2);
- **Our proposed hybrid estimator tends to do so by less** (Figures 2, 8, 9), resulting in more accurate estimation relative to $SE^*$ (Table 1);
- **Our hybrid entropy estimator outperforms other explicit semantic entropy estimators**, including plugin DSE and white-box SE, on the incorrectness detection task (Figure 3);
- **Alphabet size estimation can perform surprisingly well at incorrectness detection**, and two alphabet size estimators (hybrid alphabet size and the related $U_{EigV}$) perform closely to KLE, with all three including the top position in their 95% rank CIs ([1,4], [1, 3], and [1, 3], respectively). This is slightly updated from our original statement, which reported hybrid alphabet size and KLE sharing the top rank interval and $U_{EigV}$ lower, but it strengthens our claim that alphabet size estimation can be a simple, interpretable, and surprisingly effective UQ method in its own right.

We would like to reiterate our appreciation for the reviewers’ time and consideration. We have found their insight invaluable in this process.

---

### Meta-Review · Area_Chair_3F7r · 2025-12-28

**Summary:**

Authors propose a better (less biased) estimator for semantic-alphabet-size in the semantic-entropy-related methods for UQ estimation in NLG tasks. They show empirically that their semantic-alphabet-size is competitive with some of the SOTA methods.

**Reviewer Concerns:**

1, Lack of principled theoretical justification for the hybrid estimator design
2, Unclear conditions under which Good–Turing is valid for LLM semantic sampling
3, No systematic sensitivity analysis of semantic clustering

**Reviewer Scores:**

remain unchaged

---

### Decision · Program_Chairs · 2026-01-26

Accept (Poster)